# The Effects of Flavonoids in Cardiovascular Diseases

**DOI:** 10.3390/molecules25184320

**Published:** 2020-09-21

**Authors:** Lorena Ciumărnean, Mircea Vasile Milaciu, Octavia Runcan, Ștefan Cristian Vesa, Andreea Liana Răchișan, Vasile Negrean, Mirela-Georgiana Perné, Valer Ioan Donca, Teodora-Gabriela Alexescu, Ioana Para, Gabriela Dogaru

**Affiliations:** 1Department 5—Internal Medicine, 4th Medical Clinic, Faculty of Medicine, “Iuliu Haţieganu” University of Medicine and Pharmacy, 400015 Cluj-Napoca, Romania; lorena_ciumarnean@yahoo.com (L.C.); mircea_milaciu@yahoo.com (M.V.M.); albmirela@yahoo.ro (M.-G.P.); Vasile.Negrean@umfcluj.ro (V.N.); teodora.alexescu@gmail.com (T.-G.A.); ioana.para@yahoo.com (I.P.); 2Regional Institute of Gastroenterology and Hepatology ‘Octavian Fodor’ Cluj-Napoca, 400162 Cluj-Napoca, Romania; runcan.octavia@yahoo.ro; 3Department 2—Functional Sciences, Discipline of Pharmacology, Toxicology and Clinical Pharmacology, Faculty of Medicine, “Iuliu Haţieganu” University of Medicine and Pharmacy, 400337 Cluj-Napoca, Romania; 4Department of Pediatrics, “Iuliu Hațieganu” University of Medicine and Pharmacy, 400177 Cluj-Napoca, Romania; 5Department of Geriatrics-Gerontology, “Iuliu Haţieganu” University of Medicine and Pharmacy, 400139 Cluj-Napoca, Romania; valerdonca@gmail.com; 6Department of Physical Medicine & Rehabilitation, “Iuliu Hațieganu” University of Medicine and Pharmacy, Louis Pasteur Street 6, 400349 Cluj-Napoca, Romania; dogarugabrielaumf@gmail.com

**Keywords:** flavonoids, polyphenols, cardiovascular disease

## Abstract

Flavonoids are metabolites of plants and fungus. Flavonoid research has been paid special attention to in recent times after the observation of their beneficial effects on the cardiovascular system. These favorable effects are exerted by flavonoids mainly due to their antioxidant properties, which result from the ability to decrease the oxidation of low-density lipoproteins, thus improving the lipid profiles. The other positive effect exerted on the cardiovascular system is the ability of flavonoids to produce vasodilation and regulate the apoptotic processes in the endothelium. Researchers suggested that these effects, including their anti-inflammatory function, are consequences of flavonoids’ potent antioxidant properties, but recent studies have shown multiple signaling pathways linked to them, thus suggesting that there are more mechanisms involved in the beneficial effect of the flavonoids on the human body. This review aims to present the latest data on the classification of these substances, their main mechanisms of action in the human body, and the beneficial effects on the physiological and pathological status of the cardiovascular system.

## 1. Introduction

Cardiovascular disease (CVD) is the leading cause of global mortality and morbidity. There has been an increase in mortality among women with cardiovascular disease, which in recent years has exceeded the number of deaths from breast cancer [1]. The most commonly cited non-modifiable cardiovascular risk factors are age and gender. The incidence of cardiovascular events increases with age, on the one hand due to the increase in plasma cholesterol, and on the other hand due to increased arterial stiffness and peripheral vascular resistance [2]. The gender-related risk of CVD depends on age, so the incidence of CVD in men aged under 50 is 3–5 times higher than in women. Over the age of 50, there is a significant increase in the incidence of CVD among women. Other risk factors mentioned in the literature are genetic factors, sedentary lifestyle, obesity, smoking, high blood pressure, diabetes, and dyslipidemia [3].

Flavonoids are an important class of natural substances, with polyphenolic structure, characterized by a general structure consisting of two benzene rings (two phenyl rings and a heterocyclic ring). The three components lead to the formation of the basic structure of flavonoids, that is to say, the flavan nucleus [4]. Flavonoids from plants are synthesized from phenylalanine, which via phenylpropanol will be transformed into 4-coumarate coenzyme A. This process leads to the formation of chalcone, which has structure similar to that of flavonoids and represents a precursor for numerous flavonoids [5,6]. Depending on the enzymes that act on chalcone and on the plant it derives from, two large classes of flavonoids will result: 2-phenylchromen or 3-phenylchromen [7]. The first class includes flavones, flavonols, flavanones, flavan-3-ols, and anthocyanidins, and from the second group, isoflavonoids [8,9].

Flavonoids can be found in fruits, vegetables, nuts, seeds, in coffee, wine, or tea, with significant antioxidant effects associated with various pathologies such as cancer, atherosclerosis, Alzheimer’s disease, etc. [10,11]. At the plant level, they are responsible for the defense against oxidative stress, they act as UV filters and protect plants from different biotic and abiotic stresses, function as signal molecules, detoxifying agents and antimicrobial defensive compounds, and they are responsible for the color of fruits or flowers and their aroma. Polyphenols are the most abundant antioxidants in the diet [12]. Their intake is 10 times higher than that of vitamin C and 20 times higher than vitamin E or carotenoids. Through their antioxidant, anti-inflammatory, anti-carcinogenic, and antimutagenic action, they find use in various nutritional, pharmaceutical, medicinal, and cosmetic applications [13]. A large variety of flavonoids was identified and grouped based on chemical structure and degree of oxidation [14].

The amount and composition of polyphenols in plants is different depending on the species, age, portion of the plant, method of cultivation, storage, and geographical distribution [15]. Flavonoids play various roles in the biological activities of plants, animals and bacteria. Environmental factors like soil type, exposure to solar light, rainfall, and the number of fruits growing in a tree directly influence the concentration of polyphenols in plants [7].

A quick review of the Web of Science Database using the keywords ″flavonoids AND cardiovascular disease″ retrieves 139 papers for 2020, from which we selected the ones reporting plant names/flavonoid names in the title. We have reviewed these papers and synthesized them into the table below (Table 1), in order to show the recent trends in flavonoid research.

The phenols bioavailability depends on numerous parameters such as the source from which they are administered, whether they are pharmaceutical or nutritional products, digestion, the degree of functionality of the intestinal microflora, absorption, and metabolism [16]. Most polyphenols are present in the form of glycosides. In order to be absorbed, glycosides must be subjected to the action of intestinal hydrolases, as they are hydrophilic and cannot be absorbed by passive diffusion. In contrast, aglycone is absorbed much more easily by passive transport as they are highly hydrophobic [17]. After being modified by different pathways, the rapidly absorbed flavonoids move past enterocytes and reach the liver, where they will be subjected to other metabolic processes, such as glucuronidation, methylation etc. [18] From the liver, part of the metabolites will be distributed in the blood and the other part will be secreted into the bile to participate in the enterohepatic circuit [19].

Certain polyphenols such as soy isoflavone are well absorbed through the intestinal membrane, while others, such as proanthocyanidin found in wine and cocoa or thearubigin, the main polyphenol in black tea, are poorly absorbed [20,21].

In recent years, flavonoids have received a great deal of attention from specialists for the variety of potential benefits they offer. It is a complex and long-term study due to the heterogeneity of molecular structures, but numerous studies have suggested that dietary polyphenols may be beneficial as adjunctive treatment for the prevention and treatment of chronic inflammatory diseases [22,23]. Even though nowadays there are many plants known to possess a high concentration of flavonoids, maybe the best known are the vegetables and fruits extensively used as daily food. The table below presents a series of beneficial effects and composition of foods rich in flavonoids (Table 2) [23,24,25,26,27].

## 2. Classification of Flavonoids

Natural polyphenolic compound flavonoids are classified into six main subgroups, namely: flavones, flavanones, flavan-3-ols, flavanols, anthocyanidins, and isoflavones [28,29].

### 2.1. Flavones

Flavones are found in foods such as celery, garlic and chamomile tea, being rich in luteolin [30]. Among the beneficial effects of luteolin, observed in various studies, are the ability to lower blood pressure in hypertensive rats, improve vasodilation in aortic rings, increase the accumulation of cAMP by inhibiting cAMP-specific phosphodiesterase [31]. By activating the cAMP/PKA cascade, nitric oxide levels increase in endothelial cells by activating endothelial nitric oxide synthase. This induces vascular relaxation through nitric oxide, a mechanism regulated by calcium and potassium channels [32].

### 2.2. Flavonols

Flavonols are found in onions, broccoli, tea and fruit, and are represented by glycosides, namely quercetin and kaempferol [4].

#### 2.2.1. Quercetin

Quercetin exerts its antihypertensive effect by the ability to improve endothelial function, to modulate the renin-angiotensin-aldosterone system (by modulating the mechanism of contraction of smooth muscles in blood vessels) [33], produces vasodilation at renal level that is protein kinase C-dependent, and lowers blood pressure in patients with diabetes or metabolic syndrome [34,35]. Quercetin also decreases oxidative stress in the heart and kidneys [36].

#### 2.2.2. Kaempferol

Kaempferol, a compound of foods such as broccoli, green tea, strawberries, and beans [30], has antihypertensive properties, effects manifested by the action of endothelial nitric oxide [37]. Besides the antihypertensive effect, kaempferol has the ability to reduce albuminuria and proteinuria, being considered a potential candidate in the improvement of these two situations [38].

### 2.3. Flavan-3-ols

Flavan-3-ols include monomers such as catechin, gallocatechin, epicatechin, and oligomers (proanthocyanidin) [30]. Catechin monomers are found in the form of aglycones (part of a glycoside without carbohydrate content) in apples, pears, cocoa, tea, and grape-based products. It has been shown that catechins have beneficial effects on vascular function and have a cardioprotective effect. Moreover, studies have shown that they have the ability to reduce both systolic and diastolic blood pressure [39,40].

#### 2.3.1. Epicatechin

Epicatechin presents antihypertensive action. Eating epicatechin-rich chocolate helps decrease systolic blood pressure by 4.2 mmHg and diastolic blood pressure by 2.1 mmHg. It has recently been shown to reduce myocardial rigidity in rats with hypertrophic cardiomyopathy [41].

#### 2.3.2. Epigallocatechin-3-gallate

Epigallocatechin-3-gallate abundant in green tea, has anti-inflammatory, antioxidant, and anti-atherogenic action [42].

### 2.4. Flavanones

Flavanones: the main representatives of this class are naringenin and hesperetin, being predominantly found in citrus fruits and fruit peels [43]. They demonstrate antioxidant properties by blocking the activity of free radicals [44].

#### 2.4.1. Naringenin

Naringenin has the following primary effects: reduces blood pressure, modulates nitric oxide levels, and protects against endothelial dysfunction [45,46].

#### 2.4.2. Hesperetin

Hesperetin is a dietary flavanone present in citrus fruits [30], which is rapidly absorbed in the intestine, the resulting metabolites being responsible for the antihypertensive effect. They can also reduce the progression of atheroma plaque through their anti-inflammatory activity. In addition to this, the antioxidant effect of hesperetin helps to increase the amount of nitric oxide and reduces the amount of calcium ions, thus producing a smooth muscle relaxation in the blood vessels [47,48].

### 2.5. Anthocyanidins

Anthocyanidins are soluble pigments responsible for the color of fruits and vegetables, especially red, blue or purple fruits (forest fruits, black currants) [30]. The beneficial effects on the cardiovascular system are exerted by the endothelium-dependent vasodilation and by reducing the risk of acute myocardial infarction [49].

### 2.6. Isoflavones

Isoflavones, when found in soy they have a structural similarity to mammalian estrogens, so they are thought to act as estrogen receptor agonists. Among the main representatives of these classes are diadzein and genistein [50].

#### 2.6.1. Diadzein

Diadzein exerts its effect mainly by diminishing the damage caused by oxidative stress but also by increasing nitric oxide synthesis, by reducing LDL oxidation or by increasing the production of prostaglandins [51].

#### 2.6.2. Genistein

Genistein has an important antihypertensive effect [52].

## 3. The Flavonoid’s Mechanism of Action in Different Pathological States

Flavonoids have many beneficial effects on health, the most important being the antioxidant, anti-inflammatory, antiplatelet, antihypertensive, and anti-ischemic effects [53].

### 3.1. Flavonoids and Their Antiplatelet Effect

Studies have shown that excessive platelet activation is closely linked to various chronic diseases, the most commonly cited being high blood pressure, diabetes and various other vascular diseases. Due to the high content of adhesion proteins in the granules, their excessive activation has direct implications in the progression of different thrombotic diseases [54,55]. One of the key events in platelet activation is the conversion of arachidonic acid (under the action of cyclooxygenase 1) to thromboxane A2 [14].

The many beneficial effects of flavonoids include the ability to interfere with lipid metabolism, decrease platelet adhesion, and improve endothelial function [14,56]. The main mechanism by which flavonoids prevent platelet aggregation is based on their involvement in arachidonic acid metabolism. As mentioned above, thromboxane A2 is the main compound resulting from arachidonic acid metabolism containing receptors on the surface for the induction of platelet aggregation. Studies have shown that flavonoids exhibit antagonism to thromboxane A2 receptors, suggesting that flavonoids, by indirectly inhibiting cyclooxygenase 1, reduce thromboxane A2 levels [57]. Non-glucuronidated genistein and diadzene were studied and it was shown that the two can bind to the receptors on the surface of thromboxane A2, thus interfering with platelet aggregation [58].

Besides their intervention in arachidonic acid metabolism, flavonoids also reduce platelet aggregation by the changes produced in the metabolism of collagen, knowing that collagen has a significant influence in the first phase of platelet aggregation. Oxidative stress can activate collagen-induced platelet aggregation, activating the inositol pathway and increasing intracellular calcium levels. It has been shown that certain flavonoids such as quercetin, catechin or kaempferol can reduce oxidative stress, blocking NADPH-oxidase [54,59].

Plant extracts have different mechanisms by which they can exert their antiplatelet effect, the most obvious ones being as follows: blocking the rise of intracellular calcium, which is a key mechanism for the inhibition of thrombus development and cytoskeletal reorganization, inhibiting thrombocyte secretion, blocking the formation of thromboxane, increasing the intracellular concentration of cAMP and cGMP, inhibiting the cleavage of phosphatidylinositol or protein C, and inhibition of phospholipase C and platelet activation factor. Flavonoids have been shown to be effective in inhibiting various enzymes such as phospholipase A, certain tyrosine kinases, lipoxygenases, cyclooxygenases, or phosphodiesterase [60].

In vitro studies have shown that certain flavonoids such as quercetin or catechin influence the level of nitric oxide, increasing it and decreasing the expression of the glycoprotein IIB/IIIa complex, thus blocking the platelet aggregation in selected cases [61].

### 3.2. Flavonoids and Their Antioxidant Effects

The human body is able to naturally maintain the balance between oxidants and antioxidants through antioxidant defense systems. The main oxidants are the reactive oxygen species: hydroxyl group (-OH), superoxide and peroxynitrite. Under pathological conditions, antioxidant defense systems are outdated, and thus reactive oxygen species are generated [62,63].

Of all the beneficial effects of flavonoids on health, the most studied is their antioxidant effect. The antioxidant potential depends on the molecular configuration, the position and number of hydroxyl groups. Flavonoids, by their ability to donate electrons to peroxynitrite, hydroxyl, and peroxyl radicals, reduce the levels of reactive oxygen and nitrogen species, due to the formation of stable flavonoid radicals and the stabilization of the aforementioned radicals [4,64].

The antioxidant effect of flavonoids is achieved by three mechanisms: 1) by eliminating reactive oxygen species, 2) by preventing the production of reactive oxygen species, secondary to the interaction of flavonoids with enzymes that control the production of free radicals, or 3) by increasing the protection of antioxidant systems. The antioxidant effect is achieved by a single mechanism or by involving several mechanisms at the same time, for example flavonoids can block the enzymes that lead to the formation of reactive oxygen species and at the same time can also directly eliminate them [65]. Specific enzymes on which flavonoids act are glutathione S-transferase, NADH oxidase, microsomal monooxygenase, and mitochondrial succinoxidase [66,67,68].

Together with the formation of reactive oxygen species, there is also an increase in the levels of free metal ions. By their low redox potentials, flavonoids have the ability to chelate these metal ions, thus reducing the formation of free radicals. The flavonoid with the highest chelating capacity of metal ions is quercetin [67].

Under oxidative stress, flavonoids additionally achieve lipid protection from the peroxidation process. Their effect on lipid oxidation is due to the interaction of flavonoids with nonpolar compounds in the hydrophobic portion of the membrane. In the hydrophobic region it blocks the access of oxidants, thus protecting the membrane structure. In vitro, this protective effect was observed mainly with epicatechin. Epicatechin also has a powerful effect of eliminating free radicals [69].

Bacterial endotoxins as well as inflammatory cytokines stimulate nitric oxide synthase expression leading to nitric oxide synthesis and subsequent oxidative injury. The action of polyphenols on the enzymes involved in the metabolism of arachidonic acid (cyclooxygenase, lipoxygenase, nitric oxide synthetase), allows the inhibition of the inflammatory effect by decreasing the synthesis of nitric oxide, prostaglandins or leukotrienes [70].

Certain flavonoids have been shown to be effective in reducing xanthine oxidase activity. The xanthine oxidase pathway produces metabolites that have the capacity to induce oxidative injury through the synthesis of reactive oxygen species. Carbohydrate fragments from the structure of flavonoids play an important role in their antioxidant action. Aglycones have been shown to be stronger antioxidants than the glycosides to which the aglycones belong. Some studies have shown, however, that the antioxidant activity of flavonoids decreases with the increase in the number of glycosidic moieties linked to a hydrophobic aglycone [71].

### 3.3. Flavonoids and Their Anti-Inflammatory Effects

Inflammation is a complex biological process that occurs in response to harmful stimuli, and it depends on a set of enzymes (cyclooxygenase, lipoxygenase, tyrosine kinase, phospholipase A2, protein kinase C). Studies have shown that certain flavonoids act directly on one or more of these enzymes, inhibiting them, thus having a direct effect on the regulation of inflammation [66,72,73].

Acute inflammation is known to play a physiological role of defense and healing, but in some pathological situations, the mechanisms of inflammation regulation are altered, resulting in a long-lasting inflammatory process (chronic inflammation), which disturbs homeostasis. Chronic inflammation will later lead to the development of certain pathologies such as cancer, diabetes, cardiovascular disease or neurological disease. Chronic inflammation causes tissue damage through pro-inflammatory cytokines (IL-1, IL-6, IL-8, IL-13, tumor necrosis factor α, amyloid A, C-reactive protein). They recruit immune cells and platelets, which in turn will secrete other pro-inflammatory molecules. Oxidative stress that occurs during the inflammatory process will affect cell integrity through the destruction of lipids, proteins, and nucleic acid [74].

The integrity of the immune system can be influenced by diet, medication or pollutants in the environment. Epidemiological studies have shown that diet plays one of the most important roles in combating chronic inflammation. In vitro and in vivo studies have shown the anti-inflammatory activity exerted by certain flavonoids.

Prostaglandin synthesis is one of the main stages influenced by flavonoids. In vivo studies have shown that hesperidin and diosmin can inhibit prostaglandin synthesis [75]. A number of in vitro studies have shown that certain flavonoids (bilobetine, morelloflavone, amentoflavone) as well as flavonoids in *Sophora Flavescens* work by blocking the release of arachidonic acid [76]. Their action is due to the direct inhibition of phospholipase A2 as well as phospholipase C1. Quercetin blocks the synthesis of prostaglandins, leukotrienes and thromboxanes by inhibiting the cyclooxygenase and lipoxygenase [77]. Another flavonoid with anti-inflammatory action is resveratrol. It modulates the anti-inflammatory response by inhibiting the synthesis of eicosanoids, nitric oxide (by inhibiting nitric oxide synthase) or prostaglandin, as well as by blocking the synthesis and the release of certain inflammatory mediators [78]. The tannins, luteolin and apigenin from *Cymbopogon citratus (DC) Sapf.* produce vasorelaxation, possessing also an antioxidant and anti-inflammatory potential [79]. This potential is also achieved by flavonoids from *Anchusa italica* Retz. and *Heliotropium taltalense* Phil. (mainly rutin, hesperidin, quercetin, or naringenin, pinocembrin, respectively) [80,81].

Studies have shown that certain citrus flavonoids inhibit the synthesis of various pro-inflammatory mediators, the most important being prostaglandins and thromboxanes A2 [82]. Apigenin has a potent anti-inflammatory effect, which may in the future be an alternative to current anti-inflammatory medication due to the lack of adverse effects. *Artemisia Copa*, a plant rich in flavonoids (spinacetin, penduletin, tricine, jaceosidin), has anti-inflammatory action by inhibiting nitric oxide synthase and cyclooxygenase resulting in decreased levels of prostaglandins and nitric oxide. The flavonoids in this plant have proved to be effective in blocking the activity of phospholipase A2, the most potent of these being jaceosidin [83].

It has been shown that hydroxylated flavonoids (aglycones) are effective in inhibiting the synthesis of leukotrienes, thus affecting the late phase of allergic reactions. Studies have shown that they can reduce histamine release by inhibiting basophil degranulation. Aglycones inhibit neutrophil degranulation to a certain extent, thus reducing the concentration of arachidonic acid [84]. Studies have shown the effectiveness of a number of herbs in inhibiting phosphodiesterases, thus improving chronic inflammation and allergic reactions. Aglycones have proved to be most effective in inhibiting cAMP phosphodiesterase. Biflavonoids from *ginkgo biloba* have also proved to be phosphodiesterase inhibitors in in vitro studies [85].

### 3.4. Effects of Flavonoids in Hypertension

Cardiovascular disease ranks first in the world in terms of mortality and morbidity. There are numerous risk factors for these conditions, the most important being high blood pressure, age, obesity, dyslipidemia, sedentary lifestyle, smoking, stress, and inadequate lifestyle [29].

It is well known that nitric oxide (NO) from the endothelium plays a crucial role in regulating vascular tone and blood pressure. The mechanism of action of NO is based on the activation of the cGMP-protein kinase G cascade in smooth muscle cells in vessels. Once the cascade is activated, there is stimulation of potassium channels, a process that results in membrane hyperpolarization and inhibition of intracellular calcium influx, producing vasodilation. The action of protein kinase G is based on the phosphorylation of myosin light chains, a process by which the vasoconstriction of the smooth muscles in the vessels decreases [86,87].

Flavones, a subgroup of luteolin-rich flavonoids, exert their antihypertensive effect by signaling and activating the cAMP/protein kinase A cascade, which will further activate NO synthase, the final result being the increased concentration of endothelial NO. Through this mechanism, vasodilation takes place, a process modulated by potassium and calcium channels [31,32,88].

The antihypertensive capacity of flavonols represented by kaempferol and quercetin is manifested by modulating the renin-angiotensin-aldosterone system, by improving endothelial dysfunction and by regulating the contraction of smooth muscle in vessels [33,37,38]. These mechanisms are due to their ability to activate NO-synthase 3, and ultimately resulting in higher levels of plasma NO. The improvement of endothelial function is achieved by suppressing the response of smooth muscle cells in vessels to the action of endothelin-1 [33,34].

In vitro studies have been performed based on endothelial denudation, thus demonstrating that the antihypertensive effect of quercetin and kaempferol is dependent on NO synthesized in the endothelium. This theory was proposed as a significant reduction of the vasodilator activity of the two flavonols was observed when using endothelial denudation [34,35,36].

Unlike quercetin, naringenin, a representative of the flavanone class, also manifests its vasodilatory effect in denuded endothelium due to its ability to activate potassium channels, especially those activated by calcium as well as voltage-dependent ones. The capacity of naringenin to lower blood pressure is due to both membrane hyperpolarization and relaxation of vascular smooth muscle, processes influenced by calcium-activated potassium channels [89]. This shows the usefulness of naringenin as a therapeutic agent in the treatment of hypertension [45,46]. The second representative of the flavanone class, hesperetin, produces vasodilation through its active metabolite (hesperetin-7-0-betaglucuronide) by increasing the adhesion of nitric oxide synthase and reducing the levels of nitrous oxide. Thus, there is an increase in plasma levels of NO [47]. The second mechanism involved in vasodilation is the reduction of intracellular calcium ions by blocking voltage-gated calcium channels [48].

A study in rats with hypertension showed that the antihypertensive mechanism of action of epicatechin is based on the reduction of superoxide production in the aorta and the left ventricle, but also on the increase in NO-synthase activity [90]. In addition, epicatechin also acts by decreasing arginase-2 levels, a substance that is responsible for reducing NO [39,40,41].

As mentioned above, soy isoflavones are structurally similar to human estrogen, which is why it is thought they have a beneficial effect in preventing the onset of hypertension in menopausal women. Daidzein causes vasodilation by the same mechanisms found in the case of other flavonoids, differing from them by the additional property of stimulating the production of prostaglandins [51]. Genistein exerts its antihypertensive effect by inhibiting tyrosine kinase Pyk2, the enzyme responsible for the regulation of calcium ion channels and activation of signaling pathways [52]. Another important effect of genistein is the reduction in pulmonary hypertension through its ability to reduce smooth muscle hypertrophy in pulmonary arteries [91].

### 3.5. Flavonoids and Their Antiatherogenic Effects

Atherosclerosis is a chronic, slowly progressive pathology, characterized by the accumulation of lipids in the arterial wall. It is the leading cause of death from acute myocardial infarction or stroke. The primary lesion occurs secondary to the accumulation of foam cells in the subendothelial space, cells resulting from the incorporation of cholesterol, especially LDL-c, by macrophages [92].

The presence of macrophages at this level leads to the excessive release of oxidized LDL-c molecules. Moreover, macrophages can induce an inflammatory response. The consequences of this response are increased migration and proliferation of smooth muscle cells and recruitment of other inflammatory cells, thus creating a vicious circle [93].

One of the many beneficial effects of flavonoids is their antiatherogenic effect [94]. Numerous studies have been conducted in order to explain this protective effect of flavonoids [95,96,97,98]. Moderate consumption of red wine is considered to reduce the incidence of atherosclerotic disease directly by reducing the oxidation of low-density lipoproteins and by reducing endothelial toxicity provoked by oxidized LDL molecules [95]. An experimental study showed that the consumption of procyanidin (anthocyanidin found in citrus fruits) by patients with diabetes, significantly reduces the levels of oxidized LDL-c [96]. Another property of polyphenols in grape derivatives is to lower plasma lipid levels, including postprandial lipemia. It has been observed that the consumption of red wine during the meal leads to a significant decrease in lipid hydroperoxides, a fraction considered highly atherogenic [97]. Moreover, in rabbits with hypercholesterolemia, the administration of flavonoids reduces the accumulation of lipids in the iliac artery [98].

A study on hamsters fed with a hypercholesterolemic diet revealed that chronic administration of dietary polyphenol has a positive effect on plasma triglyceride, apolipoprotein B, and cholesterol levels. In the case of hamsters with normal plasma lipid levels, acute administration of flavonoids leads to an increase in HDL-cholesterol and a significant decrease in free fatty acids, triglycerides, and apolipoprotein B [99,100].

In addition to the direct effect of flavonoids on the lipid profile, other mechanisms, such as the stimulation of NO synthesis, the antiplatelet effect, and the reduction of vascular inflammation, may further contribute to the antiatherogenic effect [101].

### 3.6. Flavonoids and Ischemia

Many studies have shown that there is an inverse relationship between flavonoid intake and the risk of acute myocardial infarction (AMI), and even the complications secondary to AMI may be diminished by a moderate intake of red wine [102,103,104]. It was shown that intake of polyphenols (quercetin, resveratrol) or foods rich in these substances (red wine or natural grape juice) may reduce cell suffering caused by myocardial ischemia and at the same time improve contractile dysfunction following heart attack. Follow-up studies found that the protection of myocardial cells following infarction occurs due to the properties of flavonoids to reduce oxidative stress and increase NO levels [102,103]. Anthocyanin intake was also proven to be associated with a lower risk of myocardial infarction in young and middle-aged women [105].

Regarding brain ischemia, severe disruption of blood flow leads to a chain reaction of metabolic changes. The first change is the alteration of energy metabolism, followed by local homeostatic disturbances (especially ionic homeostasis, acid-base balance), which will stimulate an abundant secretion of excitatory amino acids. The end result of these disorders is an increase in oxidative stress that will cause neuronal death and tissue damage. A diet rich in flavonoids can reduce the incidence of ischemic strokes due to the protective effects of reducing blood pressure, lipid oxidation, and improving endothelial function. Moreover, in a study carried out on experimental models it was observed that the volume of damaged tissue following ischemia may be reduced by the administration of resveratrol, possibly due to diminished lipid oxidation. The number of neurons affected by ischemia may be reduced by polyphenols found in grapes [106].

Analyzes performed in the study performed on rats, showed that the main mechanism by which polyphenols positively influence the evolution of an ischemic stroke is by moderating the release of excited amino acids, mainly glutamate and aspartate. It has been observed that flavonoids have no effect on oxidative stress or local metabolism [107].

Although there is a growing body of evidence that suggests the neuroprotective effects of flavonoids, Rees et al. [10] have summed up that we do not fully understand the mechanisms by which flavonoids exert their action on the cerebrovascular health, mainly due to the heterogeneity of the studies existing in the literature.

To sum up, we synthesized all the main beneficial actions of flavonoids described above in the table below (Table 3), underlining the main mechanisms by which they exert protective effects in the human body.

## 4. Flavonoid Bioavailability–Can It Be Enhanced?

Since its beginnings (in the 1930s), the flavonoid research has grown exponentially [108]. Two of the most challenging attempts made were to understand their natural/synthetic availability and to enhance their bioavailability [109]. The low availability of flavonoids is well-documented, but recent efforts are done to ameliorate this concern. Recent evidence that suggested that the gut and its microbiome play an important role in producing phenolic metabolites [108], with prebiotic properties and action as modulators of microbiota [110]. It has become clear that flavonoids inhibit gastrointestinal inflammation, with their metabolites being modulators of the gut immune function [111]. Thus, efforts to improve the bioavailability of flavonoids focus mainly on increasing their intestinal absorption [108,109]. The research has evolved from classical drug absorption enhancers like the borneol/methanol mixture [112] to complicated nano-delivery systems, and even nanoantioxidant flavonoid carriers [113]. In the meantime, consistent progress is also being made in the field of flavonoid extraction [114]. All these advances will offer a better bioavailability for flavonoids, enhancing their favourable effects on the human health.

## 5. Conclusions

Flavonoids have an important involvement in preventing cardiovascular disease, mainly due to their antiatherogenic, antithrombotic, and antioxidant properties. In vitro and in vivo studies have shown that flavonoids can modulate the activity of numerous inflammatory mediators, and can also inhibit immune cells, thus representing an alternative in the development of new anti-inflammatory drugs. Further studies are needed in order to fully understand all the mechanisms by which the flavonoids interact with the human body homeostasis.

## Figures and Tables

**Table 1 molecules-25-04320-t001:** Selection of recently published papers discussing the relationship flavonoids–cardiovascular disease.

Plant	Flavonoids	Action on the Cardiovascular System	Targeted Conditions
*Polygonum minus (Persicaria minor)*	Myricetin, quercetin, methyl-flavonol	Antioxidant, anti-inflammatory	Atherosclerosis, hypertension, ischemic heart disease
*Lentil (Lens culinaris Medik.)*	Quercetin, kaempferol	Anticoagulant, anti-platelet	Cardiovascular diseases associated with hyperactivation of platelets
*Ajuga Iva* (L.)	Naringein, apigenin-7-*O*-neohesperidoside	Antioxidant, anti-inflammatory anti-hypercholesterolemia	Atherosclerosis
*Cymbopogon citratus (DC) Sapf.*	Tannins, luteolin, apigenin	Vasorelaxation, antioxidant, anti-inflammatory	Hypertension
*Anchusa italica Retz.*	Rutin, hesperidin, quercetin, kaempferol, naringenin	Anti-inflammatory, antioxidant, anticoagulant	Ischemic heart disease, myocardial infarction
*Heliotropium taltalense (Phil.)*	Naringenin, pinocembrin, quercetin	Anti-inflammatory, antioxidant, vasorelaxation	Hypertension
*Equisetum Arvense* L. (*Horsetail*)	Resveratrol, apigenin, quercetin	Antioxidant, anti-inflammatory,	Hypertension, ischemic cardiac disease
*Trichosanthes kirilowii*	Luteolin	Hypolipidemic, antioxidant, anti-atherosclerotic	Ischemic cardiac disease, hyperlipidemia, hypertension
*Thai Perilla frutescens*	Cyanidins, luteolin, phenolic acids	Anti-inflammatory, antioxidant	Ischemic heart disease, hypertension
*Abelmoschus esculentus*	Quercetin	Anti-inflammatory, antioxidant, hypolipidemic,	Atherosclerosis, stroke, hypertension
*Dracocephalum moldavica* L.	Tallianine, luteolin, apigenin, diosmetin	Antioxidant	Ischemic heart disease
*Moringa oleifera Lam.*	Catechin, epicatechin, kaempferol, quercetin	Antioxidant, anti-inflammatory	Hypertension, ischemic cardiac disease
*Ephedra herb*	Epiafzelechin (flavanol), quercetin, gallocatechin, apigenin, luteolin	Diuretic, anti-inflammatory, hypotensive, antioxidant	Hypertension
*Corchorus olitorius Leaf* and *Corchorus capsularis*	Luteolin	Antioxidant, hypotensive, diuretic	Hypertension, ischemic cardiac disease

**Table 2 molecules-25-04320-t002:** Composition and beneficial effects of certain foods rich in flavonoids [23,24,25,26,27].

Food	Bioactive Compounds	Beneficial Effects on Preventing Cardiovascular Diseases	Primary Effects
Tomatoes	Phenols: phenolic acid, flavonoids, carotenoids	Improve metabolic profile (lipid, carbohydrate metabolism), increase bioavailability of nitric oxide and vascular pressure	Antioxidant, anti-inflammatory, antiplatelet, anti-atherosclerosis, anti-hypertensive, antiapoptotic
Garlic	Allicin	Lowers LDL-cholesterol levels, blood pressure, inflammatory response, oxidative stress	Antioxidant, anti-inflammatory, anti-carcinogenic, antibacterial, antiviral, antifungal, antimicrobial
Edible wild fruits	Polyphenols: procyanidin, quercetin, phenolic acid, anthocyanin, carotenoids	Lower LDL-C levels, blood pressure, body mass index, glycosylated hemoglobin and hemoglobin levels, decrease inflammatory response and oxidative stress, improve endothelial function	Antioxidant, Anti-inflammatory, anti-obesity, anti-diabetic
Apples	Lutein, carotenoids, Antioxidant: phlorizin, quercetin, catechin, procyanidin, epicatechin	Improve lipid profile, lower blood pressure, pro-inflammatory cytokines, lipid oxidation, oxidative stress, blood glucose, increase nitric oxide	Antioxidant, antifungal, antioxidant, antifungal, anti-proliferative
Broccoli	Lutein, zeaxanthin, B-carotene, flavonoids	Improves lipid and carbohydrate profile, reduces pro-inflammatory cytokines and markers of oxidative stress	Antioxidant, anti-inflammatory, anti-carcinogenic
Cocoa	Phytochemicals: methylxanthine, proantho-cyanidin, theobromine	Improves insulin sensitivity, lipid profile, reduces blood pressure, inflammatory response and oxidative stress, improves endothelial function	Antioxidant, anti-inflammatory, hypoglycemic, antiplatelet, anti-hypertensive
Grapes	Polyphenols: resveratrol, carotenoids, flavonoids	Improve lipid profile and carbohydrate metabolism, reduce pro-inflammatory cytokines and oxidative stress, increase nitric oxide	Antioxidant, anti-inflammatory, antihypertensive, anti-diabetic
Olives	Phenolic compounds, hydroxy-tyrosol, oleuropein, polyphenols, flavonoids, theanine, quercetin	Improve lipid profile, lower blood pressure, body mass index, inflammatory response and oxidative stress	Antioxidant, anti-inflammatory, anti-hypertensive, anti-obesity

**Table 3 molecules-25-04320-t003:** Main beneficial effects of flavonoids on health and mechanisms of action.

Beneficial Effect	Specific Mechanisms
Antiplatelet	Blocking excessive platelet activation, decrease platelet adhesion [57,59,60]
Antioxidant	Formation of stable flavonoid radicals, elimination of reactive oxygen species, increasing the protection of antioxidant systems [64,65,67,68]
Anti-inflammatory	Inhibition of prostaglandin synthesis, inhibition of nitric oxide synthase, inhibition of phosphodiesterases [75,76,77,78,82]
Antiatherogenic	Reduce the oxidation of low-density lipoproteins, lower plasma lipid levels [95,96,97,98,99,100,101]
Anti-hypertensive	Modulate the renin-angiotensin-aldosterone system, increase the concentration of endothelial nitric oxide [32,33,34,88,89,90]
Anti-ischemic	Reduce cell suffering caused by myocardial or brain ischemia, increase the concentration of endothelial nitric oxide [102,103,104,105,106,107]

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
