# Peer review of "The Effects of Flavonoids in Cardiovascular Diseases"

_molecules, 2020, doi:10.3390/molecules25184320_

Round 1

Reviewer 1 Report

The review is interesting, but there are some things that need to be revised.

Author Response

The authors thank the reviewer for taking the time to evaluate our work and for the valuable observations.

Response to reviewer # 1

  1. ”In terms of the description of the mechanisms of each disease, it is too lengthy.”

Thank you for your observation. We acknowledge that we detailed too much some physiopathology of certain conditions/processes in chapter 3. We proceeded to re-write or abandon some sections, as follows:

  • In 3.1., we erased the text from line 170-177, because it described the role of the platelets, which should be already known by any researcher; also, we erased from line 181 till line 190 ”The occurrence … arachidonic acid.” (it is the mechanism of thrombus formation) and re-wrote the phrase from line 190-191 to ”One of the key events in platelet activation is the conversion of arachidonic acid (under the action of cyclooxygenase 1), to thromboxane A2 [14].”
  • In 3.2., we re-wrote the first part, from line 220 till 232 („Oxidative stress … are generated.”) to a more concise paragraph, as so: ”The human body is able to naturally maintain the balance between oxidants and antioxidants through antioxidant defense systems. The main oxidants are the reactive oxygen species: hydroxyl group (-OH), superoxide and peroxynitrite. Under pathological conditions, antioxidant defense systems are outdated and thus reactive oxygen species are generated [75, 76].”
  • In 3.3., we re-wrote the first part (from line 269-281: ”Inflammation …cytokines”), as follows: ”Inflammation is a complex biological process that occurs in response to harmful stimuli, and it depends on a set of enzymes (cyclooxygenase, lipoxygenase, tyrosine kinase, phospholipase A2, protein kinase C). Studies have shown that certain flavonoids act directly on one or more of these enzymes, inhibiting them, thus having a direct effect on the regulation of inflammation [16, 84, 85].” Also, we have erased the phrase from line 295-297 (”The release …thromboxanes [70].”), because it was previously described.
  • In 3.4., we erased the phrase from line 328-329 (”High blood pressure … death”), because it was unnecessary.
  • In 3.5., we erased the phrase describing the formation of atheroma, because it was also unnecessary (from line 378 till 383; ”This results… extracellular matrix production [23].”)
  • In 3.6., we have re-wrote the whole first paragraph (from line 408 till 417; ”Animal studies … NO levels [95, 96].”), to: ” Many studies have shown that there is an inverse relationship between flavonoid intake and the risk of acute myocardial infarction (AMI), and even the complications secondary to AMI may be diminished by a moderate intake of red wine [110, 111, 112]. It was shown that intake of polyphenols (quercetin, resveratrol) or foods rich in these substances (red wine or natural grape juice) may reduce cell suffering caused by myocardial ischemia and at the same time improve contractile dysfunction following heart attack. Follow-up studies found that the protection of myocardial cells following infarction occurs due to the properties of flavonoids to reduce oxidative stress and increase NO levels [110, 111]. Anthocyanin intake was also proven to be associated with a lower risk of myocardial infarction in young and middle-aged women [113].” We have emphasized better the proven relationship lower risk of AMI and flavonoids, finding and citing the article published in 2013 in Circulation describing the favorable effect of anthocyanin (this is also important for your observation number 3).
  • To make it easier for our readers to sum up, we added a concise table in the last part of Chapter 3, right before the Chapter 4, as follows:

”To sum up, we synthesized all the main beneficial actions of flavonoids described above in the table below (Table 3), underlining the main mechanisms by which they exert protective effects in the human body.”

Table 3: Main beneficial effects of flavonoids on health and mechanisms of action

Beneficial effect

Specific mechanisms

Antiplatelet

Blocking excessive platelet activation, decrease platelet adhesion [70, 72, 73]

Antioxidant

Formation of stable flavonoid radicals, elimination of reactive oxygen species, increasing the protection of antioxidant systems [77-80]

Anti-inflammatory

Inhibition of prostaglandin synthesis, inhibition of nitric oxide synthase, inhibition of phosphodiesterases [87-91]

Antiatherogenic

Reduce the oxidation of low-density lipoproteins, lower plasma lipid levels [103-109]

Anti-hypertensive

Modulate the renin-angiotensin-aldosterone system, increase the concentration of endothelial nitric oxide [46-48, 97-99]

Anti-ischemic

Reduce cell suffering caused by myocardial or brain ischemia, increase the concentration of endothelial nitric oxide [110-115]

  1. ”Table 2 is superfluous because, in terms of the mechanism of effect, not only vegetables and fruits, but also other plants, such as references 73, 74, 77, etc. describe the situation.”

Thank you for your observation. We acknowledge that our tables (1 and 2) were not strong points of our review. Thus, we reconsidered the tables as follows:

  • Table 1 was eliminated completely, and changed with a new table, extremely useful in emphasizing the recent trends in flavonoid research. We erased the phrase from line 82-84 (”A study conducted … in the table below [16].”) and changed it to ” A quick review of the Web of Science Database using the keywords ″flavonoids AND cardiovascular disease″ retrieves 139 papers for 2020, from which we selected the ones reporting plant names/flavonoid names in the title. We have reviewed these papers and synthesized them into the table below (Table 1), in order to show the recent trends in flavonoid research.” The new table, with a more reader-friendly horizontal disposal is synthesizing 14 of the most important articles published in 2020 in this research field.
  • Table 2 was transformed also in a more reader-friendly landscape disposal, and we explained before the table why we have chosen to present only the most common vegetables and fruits, also properly citing the relevant literature. Thus, we added from line 106 the following text, with proper citations: ”Even though nowadays there are many plants known to possess high concentration of flavonoids, maybe the best known are the vegetables and fruits extensively used as daily food. The table below presents a series of beneficial effects and composition of foods rich in flavonoids (Table 2) [37-41].”
  1. ”The flavonoid's mechanism of action in different pathological states. The corresponding flavonoid species in 3.1-3.6 need to be supplemented (follow 2 Classification of flavonoids).”

Thank you for your observation. We have acted accordingly, mentioning the references cited in Chapter 2 also into Chapter 3. For the flavonoids not so strongly covered, such as anthocyanins, we have additionally searched and found an important reference published in 2013 in Circulation – we have already described above (in the response offered to your first observation): ”Anthocyanin intake was also proven to be associated with a lower risk of myocardial infarction in young and middle-aged women [113].”

In chapter 3.3., we added a new phrase (lines 274-275), as follows: ” The tannins, luteolin and apigenin from Cymbopogon citratus (DC) Sapf. produce vasorelaxation, possessing also an antioxidant and anti-inflammatory potential [19]. This potential is also achieved by flavonoids from Anchusa italica Retz. and Heliotropium taltalense Phil. (mainly rutin, hesperidin, quercetin, or naringenin, pinocembrin, respectively) [21, 21].” 

  1. Flavonoid absorption is not very good, targeting is not too strong. In this regard, the description should be strengthened to illustrate progress and the lack of references for 2020.”

Thank you for your observation. We have re-evaluated all our reference database, and observed that we had a lack of references from 2020. We wrote the review between April-June 2020; after your observation, we have re-checked Web of Science, noticing there were many relevant articles published since June-September 2020. Thus, we had the idea of synthesizing some of the most relevant recent papers into the new Table 1.

Also, your observation prompted us the idea of constructing a new, short and concise Chapter 4, tackling the flavonoid bioavailability. Thus, we constructed „4. Flavonoid bioavailability – can it be enhanced?”, as follows (lines 401-413):

” Since its beginnings (in the 1930s), the flavonoid research has grown exponentially [116]. One of the most challenging attempts were to understand their natural/synthetic availability and to enhance their bioavailability [117]. The low availability of flavonoids is well-documented, but recent efforts are done to ameliorate this concern. Recent evidence that suggested that the gut and its microbiome play an important role in producing phenolic metabolites [116], with prebiotic properties and action as modulators of microbiota [118]. It became nowadays clear that flavonoids inhibit gastrointestinal inflammation, their metabolites being modulators of the gut immune function [119]. Thus, efforts to improve flavonoids bioavailability focuses mainly in increasing their intestinal absorption [116, 117]. The research has evolved from classical drug absorption enhancers like the borneol/methanol mixture [120] to complicated nano-delivery systems and even nanoantioxidant flavonoid carriers [121]. In the meantime, consistent progress is made also in the field of flavonoid extraction [122]. All these advances will offer a better bioavailability for flavonoids, enhancing their favourable effects on the human health.”

  1. ”Subheading 2 after the secondary numeric title, need to be modified. It's easy for readers to misunderstand.”

Thank you for your observation. We renumbered the subheadings for “2. Classification of flavonoids” section

Reviewer 2 Report

Comments to the manuscript molecules-929744 "The effects of flavonoids in cardiovascular diseases".

Authors propose a review of the positive effects of plants flavonoids on the cardiovascular system. The review is well organized, complete, clearly written and easy to understand. Scientific literature was well explored and the most interesting news are well presented. Recent data on the flavonoid classification, antioxidant properties and mechanisms of action on the physiology of the human body are provided. Effects on the cardiovascular system are discussed in the light of more possible pathways linked the action of flavonoids. In my opinion, the manuscript is suitable for publication after minor editorial changes.

1) Table 1 was reported from the bibliography without modification. In my opinion, authors may slightly improve the quality of this Table by completing the scientific names of some plants where only Genus and Species names are reported. 2) Table 2 may be improved with a horizontal disposition (more readable because of the content of the long sentences) and by adding one column with the main literature citation (just numbers) regarding the food sources of flavonoids. 3) Table 2: please check the expression "Forest fruits" may be changed with "small fruits" or "edible wild fruits"; 4) A new Table 3 may be built in order to synthesize the last long part of the review on "The flavonoid's mechanism of action in different pathological states". For every effect (antiplatelet, antioxidant, anti-inflammatory, hypertension, antiatherogenic, ischemia) a shortlist of specific mechanisms may be indicated and obviously the references to the literature.

Author Response

Response to reviewer # 2

The authors thank the reviewer for taking the time to evaluate our work and for the valuable observations.

  1. ”Table 1 was reported from the bibliography without modification. In my opinion, authors may slightly improve the quality of this Table by completing the scientific names of some plants where only Genus and Species names are reported.”

Thank you for your observation. We have reviewed Table 1, and decided to erase it from our paper, because the studied plants from that particular study are not among the ones that were given special attention recently. Thus, we erased the phrase from row 80 - ”A study conducted in 2010 selected 40 medicinal plants to assess the amount of phenols they contain. The main results of that study (arranged from the biggest to the lowest phenol content) are shown in the table below [16].” and replaced it with the following phrase: ”A quick review of the Web of Science Database using the keywords ″flavonoids AND cardiovascular disease″ retrieves 139 papers for 2020, from which we selected the ones reporting plant names/flavonoid names in the title. We have reviewed these papers and synthesized them into the table below (Table 1), in order to show the recent trends in flavonoid research”. Also, after this phrase, we inserted a new table 1 (Table 1), which summarizes the latest research/review articles in this domain (all from 2020, including many articles which were published after the original preparation of our manuscript in June 2020), published in Web of Science indexed journals. We have updated the references accordingly.

  1. Table 2 may be improved with a horizontal disposition (more readable because of the content of the long sentences) and by adding one column with the main literature citation (just numbers) regarding the food sources of flavonoids.”

Thank you for your observation. We have taken into consideration your suggestion, and changed the table distribution. Because of the multiple papers from which we gathered the information provided in this table, we have put the 5 references together, in the phrase before the table: ”The table below presents a series of beneficial effects and composition of foods rich in flavonoids (Table 2) [37-41].” and in the title of the table: ”Table 2: Composition and beneficial effects of certain foods rich in flavonoids [37-41]”

  1. Table 2: please check the expression "Forest fruits" may be changed with "small fruits" or "edible wild fruits"”

Thank you for your observation. We acknowledged your suggestion, changing in the table "Forest fruits" with "edible wild fruits".

  1. A new Table 3 may be built in order to synthesize the last long part of the review on "The flavonoid's mechanism of action in different pathological states". For every effect (antiplatelet, antioxidant, anti-inflammatory, hypertension, antiatherogenic, ischemia) a shortlist of specific mechanisms may be indicated and obviously the references to the literature.”

Thank you for your observation. We have valued your suggestion, creating a new Table 3, at the end of the main core of the review, right before the newly inserted Chapter 4 (”Flavonoid bioavailability – can it be enhanced?”). The table is easy-to-read and highlights the specific mechanisms by which flavonoids exert their protective actions on the human health.

Table 3: Main beneficial effects of flavonoids on health and mechanisms of action

Beneficial effect

Specific mechanisms

Antiplatelet

Blocking excessive platelet activation, decrease platelet adhesion [70, 72, 73]

Antioxidant

Formation of stable flavonoid radicals, elimination of reactive oxygen species, increasing the protection of antioxidant systems [77-80]

Anti-inflammatory

Inhibition of prostaglandin synthesis, inhibition of nitric oxide synthase, inhibition of phosphodiesterases [87-91]

Antiatherogenic

Reduce the oxidation of low-density lipoproteins, lower plasma lipid levels [103-109]

Anti-hypertensive

Modulate the renin-angiotensin-aldosterone system, increase the concentration of endothelial nitric oxide [46-48, 97-99]

Anti-ischemic

Reduce cell suffering caused by myocardial or brain ischemia, increase the concentration of endothelial nitric oxide [110-115]

            Furthermore, we constructed a new chapter 4, which is briefly pointing that flavonoid bioavailability might be enhanced, citing recently published articles (lines 400-413):

”4. Flavonoid bioavailability – can it be enhanced?

            Since its beginnings (in the 1930s), the flavonoid research has grown exponentially [116]. One of the most challenging attempts were to understand their natural/synthetic availability and to enhance their bioavailability [117]. The low availability of flavonoids is well-documented, but recent efforts are done to ameliorate this concern. Recent evidence that suggested that the gut and its microbiome play an important role in producing phenolic metabolites [116], with prebiotic properties and action as modulators of microbiota [118]. It became nowadays clear that flavonoids inhibit gastrointestinal inflammation, their metabolites being modulators of the gut immune function [119]. Thus, efforts to improve flavonoids bioavailability focuses mainly in increasing their intestinal absorption [116, 117]. The research has evolved from classical drug absorption enhancers like the borneol/methanol mixture [120] to complicated nano-delivery systems and even nanoantioxidant flavonoid carriers [121]. In the meantime, consistent progress is made also in the field of flavonoid extraction [122]. All these advances will offer a better bioavailability for flavonoids, enhancing their favourable effects on the human health.”

Round 2

Reviewer 1 Report

The first column (Article) in Table 1 should be placed in the last column, and the second column (Article type) should be deleted in Table 1. In addition, Table 2 and Table 1 formats are as consistent as possible, for example, the last column is Article.